# Intensified visual clutter induces increased sympathetic signalling, poorer postural control, and faster torsional eye movements during visual rotation

**Tobias Wibble**[1,2]*, **Ulrika Södergård**[1], **Frank Träisk**[1,2], **Tony Pansell**[1,2]

**1** Division of Ophthalmology and Vision, Department of Clinical Neuroscience, Marianne Bernadotte Centre, Karolinska Institutet, Stockholm, Sweden, **2** St Erik Eye Hospital, Stockholm, Sweden

* tobias.wibble@ki.se

**Data Availability Statement:** All relevant data are within the manuscript and its Supporting information files.

## Abstract

Many dizzy patients express a hypersensitivity to visual motion and clutter. This study aims to investigate how exposure to rotating visual clutter affects ocular torsion, vertical skewing, body-sway, the autonomic pupillary response, and the subjective feeling of discomfort to the stimulation. Sixteen healthy subjects were exposed to 20 seconds rotational visual stimulation (72 deg/s; 50 deg visual field). Visual stimuli were comprised of black lines on a white background, presented at low and high intensity levels of visual clutter, holding 19 lines and 38 lines respectively. Ocular torsion and vertical skewing were recorded using the Chronos Eye Tracker, which also measured pupil size as a reflection of the autonomic response. Postural control was evaluated by measuring body-sway area on the Wii Balance Board. Values were compared to data retrieved 20 seconds before and after the optokinetic stimulation, as subjects viewed the stationary visual scene. The high intensity stimulus resulted in significantly higher torsional velocities. Subjects who were exposed to low intensity first exhibited higher velocities for both intensities. Both pupil size and body sway increased for the higher intensity to both the moving and stationary visual scene, and were positively correlated to torsional velocity. In conclusion, exposure to visual clutter was reflected in the eye movement response, changes in postural control, and the autonomic response. This response may hold clinical utility when assessing patients suffering from visual motion hypersensitivity, while also providing some context as to why some healthy people feel discomfort in visually cluttered surroundings.

## Introduction

Disorders of the balance system account for approximately four percent of all visits to the emergency department, causing rising medical costs and a disproportional burden of disability [1–3]. Healthy balance is maintained through a complex integration of sensory input in the form of vestibular, proprioceptive and visual signals [4]. Dizziness, or vertigo, may arise due to

**Funding:** The authors received no specific funding for this work.

**Competing interests:** The authors have declared that no competing interests exist.

a mismatch between these systems, such as the vestibular system signalling a head tilt, e.g. due to a stroke, while the surroundings remain visually straight and still [5].

The mismatch model is of particular interest to patients suffering from non-vestibular vertigo, accounting for over 50% of all cases [5, 6]. A prominent example are the group of patients suffering from *visual vertigo*, or *visual motion hypersensitivity* (VMH) [7]. The phenomenon is also known as *supermarket syndrome* due to the symptoms often being triggered by the visually cluttered environment of a supermarket [7]. The symptoms associated with VMH include severe vertigo with nausea, increased fall risk and general discomfort. They also express greater visual dependency than the average population [8]. Due to procedural limitations clinical evaluations of balance disorders often focus on the vestibular system, lacking objective tools for evaluating non-vestibular causes [1, 3, 9].

A standard procedure for testing the balance system is to evaluate eye movements through the vestibulo-ocular reflex (VOR). Eye movements are sensitive to vestibular signalling, and are well suited for clinical testing as they present reflexive movements that are entirely dependent upon the head's position and acceleration. Ocular torsion and vertical skewing are examples of the VOR, and are seen when tilting the head, due to vestibular activation, or when viewing a visual scene being rotated, producing an optokinetic response aiming to stabilize the image on the fovea [10, 11]. The eye movement response induced by vestibular activation, called ocular counter-roll, is more pronounced, producing a torsional gain of at least 10% to a static head tilt while a visual tilt yields a gain of 1–4% [12, 13]. The relation between vestibular and visually induced skewing is largely unknown as few studies have investigated its gain to visual motion.

Investigating possible biomarkers for quantifying vertigo and categorizing its severity is of great importance for clinical balance evaluations. There is evidence that an increase in visual density contained in a rotating scene is associated with increased body-sway [14]. In a recent study we have shown how the torsion-skewing ratio is sensitive to changes in visual clutter density, with the eyes exhibiting more degrees of torsion per degrees of skewing in response to increased clutter [11]. Vertiginous patients whose symptoms are triggered by visual stimuli exhibit increased postural sway to visual rotation compared to healthy controls, also reporting worsening of subjective symptoms [15]. As VMH patients describe an increased sensitivity to visual clutter [16], we argue that investigating the eye movement response to different levels of visual densities could hold important clinical utility. In order to evaluate the eye movement reflex as a possible proxy for objective balance problems it is necessary to correlate the response to postural control and balance discomfort. One possible approach to producing an objective measure of the discomfort is through evaluating the sympathetic response. One such method is to quantify the pupil size to various situations, as the pupil will dilate during times of stress [17].

The aim of this study was to investigate the effect of rotating visual clutter intensity, denoted as low and high information density, on a series of variables in healthy adults: 1) The oculomotor response of ocular torsion and vertical skewing, 2) The postural response, measured as body-sway, 3) The sympathetic response, evaluated through monitoring pupil size, and 4) The subjective sensation of discomfort, evaluated through a subjectively self-reported visual analogue scale (VAS). Our hypothesis was that higher density visual clutter would result in faster eye movements, a stronger sympathetic response, increased body-sway and subjective discomfort to the visual motion. Our second aim was to explore the correlation of the eye movement parameters to changes in pupil size and body-sway, investigating the use of eye movement analysis as a means of indicating the level of postural imbalance to visually cluttered surroundings.

## Material and methods

### Subjects

The study was carried out with 16 healthy participants with no history of balance discomfort, disorder or drug use affecting the central nervous system. Balance discomfort was categorized as any medical diagnosis relating to vertigo, or any history of subjective balance complaints or postural instability. Subjects were divided into two age groups, young and old, and matched in terms of sex (4 men and 4 women aged 19 to 25, and 4 men and 4 women aged 49 to 65).

All subjects exhibited normal corrected visual acuity (v.a $\geq$ 1.0), stereoscopic vision (TNO $\leq$ 60"), no latent strabismus at distant larger than two prism-diopters exo- or esophoria, and normal eye motility. Vestibular integrity was assessed through Romberg's test and Head Impulse Test (HIT) for all three semi-circular canal planes. The research protocol adhered to the Declaration of Helsinki. All participants received written and oral information of the nature of the study and provided written consent upon recruitment. An ethical permit exists within the remit of the Regional Ethics Committee of Stockholm (EPN 2018-1768-31-1).

### Method

All subjects were instructed to stand with their feet roughly at shoulder width, which was delineated by markings on the floor. They were instructed to stand still for the duration of each trial, focusing on a central fixation point and to blink as usual but not keep their eyes closed. They were not to move their feet until instructed that the trial was over, and told to maintain their original posture and avoid body-sway. This allowed continuous measure of the eye movement response, pupil size as well as involuntary body-sway. The former could then be analyzed as a potential proxy for the latter measures of the objective sympathetic and postural impact caused by the visual stimuli, as well as subjective discomfort.

**Visual stimulation.** The visual stimuli were comprised of a high resolution image of black lines (0.93 deg visual angle) against a white background (Fig 1). The visual stimuli were presented at either low intensity (LI) or high intensity (HI) with regards to the amount of visual information presented (19 lines and 38 lines dispersed throughout the visual field respectively). Each subject was exposed to both the LI and HI stimuli condition, rotating 1440 degrees with a velocity of 72 deg/s for a duration of 20 seconds around a central fixation point (0.32 cm in diameter). The direction, clockwise (CW) or counterclockwise (CCW), as well as the order of exposure to the high and low intensity stimuli, were evenly randomized within each group through stratified randomization. All participants were consequently exposed to both visual intensities; the direction of these were evenly balanced in terms of age and sex between participants in order to eliminate any directional effect.

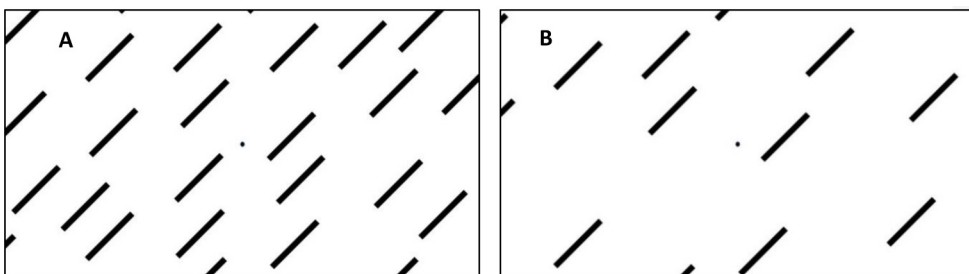

**Fig 1. A represents the high intensity stimulus (HI) holding 38 lines and B the low intensity stimulus (LI) holding 19 lines.** The black lines are 0.42cm wide and 3.25 cm long (visual angle 0.93 deg) standing at an angle of 45 degrees. The lines are centered on a fixation point, 0.32 cm in diameter. The visual scene was presented in roughly 50 degrees in the subjects' field of vision.

The stimuli were presented on a projected screen at an eye-screen distance of 2 meters in front of the test subject (res 1024 x 768; contrast 2000:1; update frequency 60Hz). The room was covered in dark grey cloth to minimize peripheral visual clues. The stimulus was initially presented at rest for 20s, after which it would rotate for 20s and finally once more be presented at rest for 20s. Each subject thusly performed two trials with a five minute break between them. In order to ascertain comparable lighting conditions a photometer (universal photometer model S4; Hagner, Solna, Sweden) was used. Candela levels per $m^2$ was measured and averaged over five points in the visual scenes: Centrally and in each corner. These trials were repeated three times for each intensity level during both the stationary and rotating phases. Averaged candela levels were compared through t-tests. There was no difference between neither visual intensity levels nor between stationary and rotating scenes.

**Eye and head recording.** The head-mounted Chronos Eye Tracker (CET; Chronos Inc, Berlin) was used to record three-dimensional eye movements, allowing for binocular recording (100 Hz) of horizontal and vertical pupil displacements with a high spatial resolution (<0.05 deg) as well as ocular torsion through rotational displacement of the iris around the pupil (<0.1 deg). The system also allowed for measuring pupil size and pupil activity, i.e. oscillations in pupil diameter size, which can be used as an indication of stress level [18].

An initial calibration was performed by letting the subjects perform a sequence of eye movements to a pattern of dots with known separations. This was done at a fixed distance between subjects' eyes and the visual pattern. This allowed for calculating pupil displacement into angular degrees by comparing the position of the eyes at each dot to the recorded eye movement. Head movements in six dimensions (3 degrees of rotation as well as translation) were quantified through a head tracking system in the Chronos mask, enabling precise measurements of head movements. This was done to ensure that subjects adhered to their instruction to not move their heads. An unwarranted head movement was defined as any shift of head position away from head straight into head roll. All head positions were plotted on a graph and visually inspected for movements deviating from the head straight position, with any movement measuring above the vestibular velocity of 1 deg/s being grounds for rejection. This was deemed to be within the vestibular threshold for angular accelerations [19]. No trials needed to be repeated due to excessive head movements.

**Postural balance recording.** One objective scale to measure body sway that is readily available and easy to use is the Wii balance-board (WBB) [20]. Postural stability was recorded using the WBB at an average frequency of 99.1 Hz (WBB; Nintendo, Kyoto, Japan). This sampling rate was gathered through multiple 60 second trials, measured with an in-built stop watch in the WBB software, and by dividing the number of frames with the time in seconds. This yielded a rate that was consistently 99.1 Hz. The WBB had four scales, one in each corner, measuring the displacement of mass through calculating the relative center of pressure (COP). In-house software had been developed for translating the WBB data for the separate scales into a measurement of weight displacement in two dimensions, horizontal by vertical, to an indexed value. This allowed for calculating postural displacement, as measured by the area in which the COP was repositioned.

After having performed all tests, the question of fatigue as a possible confounder to the body-sway arose. While studies have shown that body-sway during quiet standing is relatively insensitive to fatigue, even after fatiguing exercise [21], we wanted to eliminate any non-visual confounders to the test results. For this reason, 10 new controls were recruited. Body sway in these controls were gathered in the same manner as described above, but instead of viewing the scenes of visual clutter the visual scene held no black lines and was completely stationary during the full minute so as to present a neutral visual scene. As such, it could be deduced that any significant results from the initial trials were due to the visual stimuli.

**Rating of subjective discomfort.** After each trial the test person used a visual analogue scale (VAS) of 0–10 to rate the feeling of discomfort experienced during the trial. The task was phrased as follows: "*On this scale from 0 to 10, where 0 is no discomfort and 10 the highest level of discomfort imaginable, how would you rate your experience of the trial*?" The word *discomfort* was not further specified, as the subjects were all healthy and were not expected to experience any specific balance complaints. Additionally, it would not be possible to correlate the objective variables of pupil size and body-sway with any type of particular discomfort. Any change in discomfort rating can be considered induced by the change in visual intensity, as this was the only adjustable factor between trials. Three VAS were issued, segmenting the trial into Before, During and After the rotation of the stimulus. Subjects were handed each VAS separately in the aforementioned order immediately after having finished each trial.

## Analysis

The main outcomes of this study were: The vertical skewing velocity (left vertical eye position minus right vertical eye position), ocular torsion velocity (degrees/s), size of pupil area (pixels; px) and body sway area. Data analysis was made on the slow-phases of the nystagmus eye movement response, or in absence of nystagmus, the corresponding drift of eye position. Vertical skewing was either negative, left eye over right eye, or positive, right eye over left eye, depending on the direction of the stimulus.

Eye movement velocity was computed by calculating the change in degrees over time in seconds between the beginning and end of the slow phase. The slow phases were identified through visual inspection of the plotted eye movement response, and given as the period between two easily identified nystagmus beats. It is known that the torsion-skewing ratio is positively influenced by an increase in visual clutter during short visual rotations [11]. In order to further elucidate how the two eye movement reflexes respond to a more prolonged stimulation they were divided into three parts during the active stimulation: Early, Middle, and Late responses, collectively referred to as *Time*. This allowed us to gain some much needed context for how skewing in particular respond to visual motion, as little is known about its response pattern. Consequently it may be revealed if either torsion or skewing would prove a more reliable potential biomarker as indicated by their predictability over time.

Baseline eye movement responses before and after the stimulation phase was inspected in order to ensure gaze stability as the subject focused on the fixation point. The Early response was collected during the first five seconds (0–5), the Middle during the middle five seconds (7.5–12.5) and the Late during the last five seconds (15–20). Body sway area and pupil sizes were collected as averages of Before, During and After the optokinetic rotation.

Body sway area was calculated by producing a scatter plot of the indexed vertical and horizontal positions of each subject's COP. One of the most common ways of weighing postural instability is by fitting a 95% confidence ellipse over the COP plot [22]. This was done and the area of this ellipse was taken to be the body-sway area. As the perception of visual rotation is known to produce not only lateral body sway but also anteroposterior changes we considered the body-sway area to be the most inclusive method of postural quantification [23]. Additionally, pupil data was analysed for level of activity, i.e. oscillation frequency (Hz), as well as oscillation amplitude in order to present a more dynamic representation of the pupillary activity as compared to the average pupil size described previously. This was done using a predetermined peak-fit algorithm in Origin (OriginPro 2017, OriginLab, Northampton, MA). A signal baseline was found by asymmetric-least-squares-smoothing (smoothing factor 5, asymmetric factor 0.001, threshold 0.05, number of iterations 10) and all positive peaks were calculated from this baseline with its respective height being retrieved in Gaussian curve format through

automated window search (2% of raw data in height and width, 19% peak filtering threshold height). This peak-fit filter was calibrated based on the first subject's response profile. The number of peaks over the stimulation period was used to calculate the oscillation frequency in Hertz.

Statistical analysis was performed using IBM SPSS Statistics 25 for Windows for parametric statistics. Data distribution was inspected using the explore tool in SPSS. Tests of normality were performed by Shapiro-Wilk's test as well as visual inspection of stem-leaf plots to identify outliers. A repeated ANOVA model was used to analyse the interaction effects of Modality and Intensity. The order of visual presentation, low or high intensity first, was used as a within-subject factor. Alfa ($\alpha$) was set to 0.05. The Bayes factor was calculated as a complement to the parametric ANOVA, having been well established as a good alternative when working with small and skewed sample distributions. Bayesian statistics was analysed using JASP (Version 0.9.2) (JASP Team (2019)) to obtain the odds for or against the null hypothesis. The $BF_{10}$ value indicates how much more likely the alternative hypothesis $H_1$ is compared to the null hypothesis $H_0$. A Bayes factor larger than 10 indicates strong evidence for $H_1$, 3 to 10 indicates moderate evidence, 1 to 3 only anecdotal evidence for $H_1$ and values below 1 no evidence at all. 1/3 to 1 indicates anecdotal evidence for $H_0$, 1/10 to 1/3 moderate evidence and 1/30 to 1/10 strong evidence for $H_0$ [24]. For example, a $BF_{10}$ of 5.5 indicates that the data are 5.5 times more likely under the alternative hypothesis than under the null hypothesis.

Correlations were calculated between the eye movement responses, ocular torsion and vertical skewing, body-sway and pupil size, and finally subjective discomfort. The averaged values retrieved during the active rotation period were used, i.e. for the eye movement responses the Early, Middle, and Late responses were averaged. As no eye movement was expected during the static visual presentation no analysis involving torsion or skewing were done during these periods. Correlations were calculated using a linear regression model as Pearson correlation coefficient, with the correlation being described as weak, moderate or strong based on the general consideration that $r \leq 0.35$ is weak, r = 0.36 to 0.67 moderate, and $r \geq 0.68$ strong.

## Results

All subjects exhibited a stable gaze Before and After the stimulation phase as no eye movement responses were evoked. No differences in body sway was seen over time for the control group viewing the stationary visual fixation point. No significant difference between age groups were seen for torsion, skewing, body-sway or pupillary response. It is well established that pupil size decreases with age, which in this study yielded a smaller pupil size in the older participants at baseline [F(1,14) = 7.49; p = 0.016]. This did however not affect the analysis as the repeated ANOVA model controls for baseline differences when analyzing changes between different stimuli. Additionally, age yielded no significant effect as a between-subject factor [F(2,28) = 1.09; p = 0.350].

### Eye movement responses

Visual stimulation induced an eye movement response following the optokinetic rotation in all subjects, i.e. a clockwise visual rotation resulted in a clockwise ocular torsion. Fig 2 illustrates the torsion-skewing response from one subject to a high intensity visual stimulation, also showing the gaze stability at baseline prior to the stimulation phase and the gradual return to original eye position after the stimulus has ceased rotating. There was a significant decrease in torsion velocity during the stimulation period [F(2,30) = 12.63; p<0.001] (Fig 3). A repeated contrast analysis revealed a significant difference between both the Early and Middle (t(16) = 2.80; p = 0.009) as well as between the Middle and Late (t(16) = 2.22; p = 0.034) period. Within

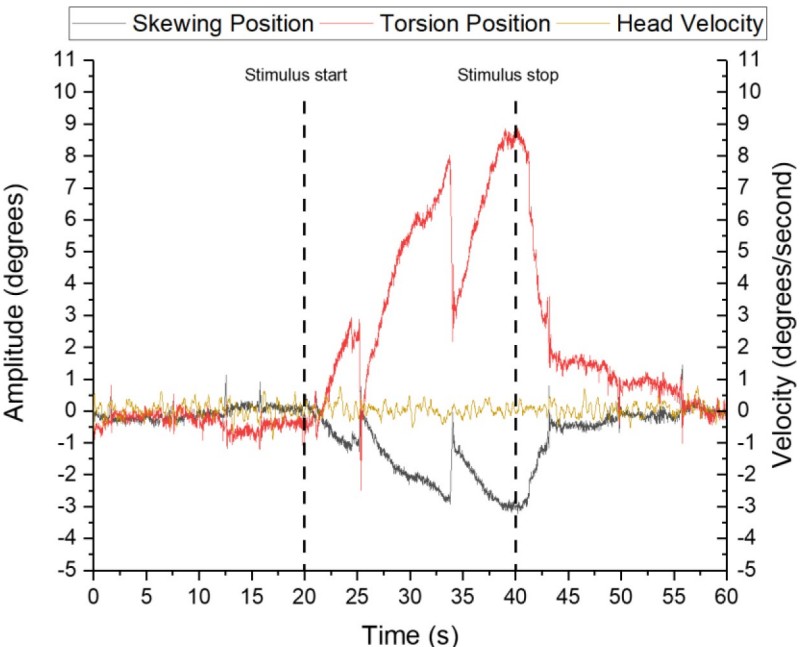

**Fig 2. Unmodified raw signal examples of ocular torsional and ocular skewing to high intensity stimulation.** Head velocity has been smoothed by Adjacent-Averaging with a Points of Window of 25. The dashed line signifies stimulus start and stop.

subject contrast revealed a significant linear declining trend [$F(1,14) = 22.43$; $p < .001$]. The response was also significantly higher for the HI stimuli compared to the LI [$F(1,15) = 11.00$; $p = .005$]. Additionally there was an interaction effect between Intensity and Intensity order [$F(1,12) = 13.39$; $p = .003$]. Subjects who were exposed to the LI first exhibited a higher torsional velocity for both intensities (Table 1). The Bayesian ANOVA comparing the effect of Time and Intensity supported the main effect of Time and the interaction effect from the ANOVA (Intensity: $BF_{10} = 2.0$; Time: $BF_{10} = 3461.0$; Intensity + Time: $BF_{10} = 14987.7$)

Skewing velocity similarly displayed a decrease during the stimulation period [$F(2,30) = 6.62$; $p = 0.004$] (Fig 3). A repeated contrast effect did however only reveal a significant

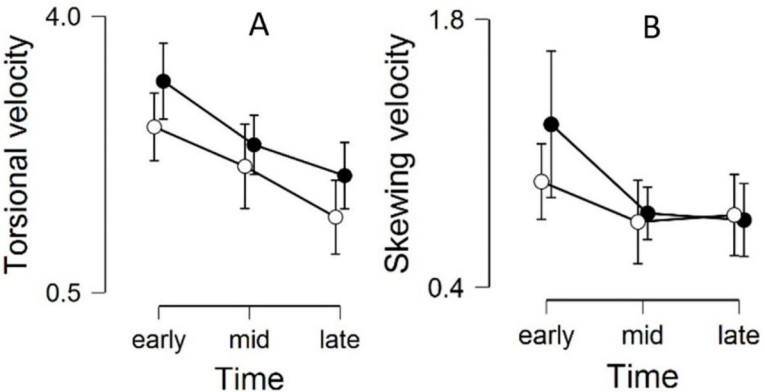

**Fig 3.** The response in A) torsional and B) skewing velocities for high intensity (black) and low intensity (white) stimuli during the early (0-5s), middle (7.5–12.5s), and late (15-20s) time period. Values are presented as means with standard deviation.

**Table 1. Torsional response based on order of presentation and visual intensity, presented as means (sd).**

| Time | Low intensity first | | High intensity first | |
|---|---|---|---|---|
| | Low intensity | High intensity | Low Intensity | High Intensity |
| Early | 2.71 (2.07) | 3.29 (1.73) | 2.49 (1.42) | 3.07 (1.67) |
| Middle | 2.21 (1.57) | 2.86 (1.69) | 1.99 (1.23) | 1.89 (1.33) |
| Late | 1.50 (0.58) | 2.74 (1.40) | 1.42 (1.37) | 1.22 (0.68) |

*Intensity* refers to the level of visual intensity presented. Early/Middle/Late refers to the time during the stimulation, which has been divided into three time slots: early (0-5s), middle (7.5–12.5s), and late (15-20s).

difference between the Early and Middle ($t$(16) = 3.16; p = 0.004) while the Middle and Late was not found to differ ($t$(16) = -0.007; p = 0.995). There was no significant effect of Intensity or Intensity order for the stimulation period. The Bayesian ANOVA supported the main effect of Time and a modest interaction effect (Intensity: $BF_{10}$ = 0.371; Time: $BF_{10}$ = 14.947; Intensity + Time: $BF_{10}$ = 5.678).

## Body sway

There was a significant difference in body sway between Before, During and After the stimulation [F(2,28) = 6.68; p = .004]. The body sway was largest Before the stimulus rotation, and smallest During. There was also a significantly larger body sway when the subject viewed the HI compared to the LI visual scene [F(1,14) = 7.25; p = .018] (Fig 4). The Bayesian ANOVA for repeated measures moderately supported the main effect of Intensity, while the effect of Time was strongly supported when included in the interaction effect with Intensity (Intensity: $BF_{10}$ = 6.932; Time: $BF_{10}$ = 2.180; Intensity + Time: $BF_{10}$ = 20.932). The postural response used to calculate the body sway area is illustrated in Fig 5, as anteroposterior and lateral changes in centre of pressure. As one can see, subjects would sway primarily in the anteroposterior direction. The same phenomenon was seen in all subjects.

## Pupillary response

The pupil responded in a similar fashion to that of body sway (see Fig 4). There was a significant difference in pupil size between Before, During and After the stimulation. The pupil was largest Before and smallest During the stimulus rotation [F(2,14) = 12.21; p = .001]. The HI stimulus yielded a significantly larger pupil size than the LI stimulus [F(1,15) = 8.21; p = .012].

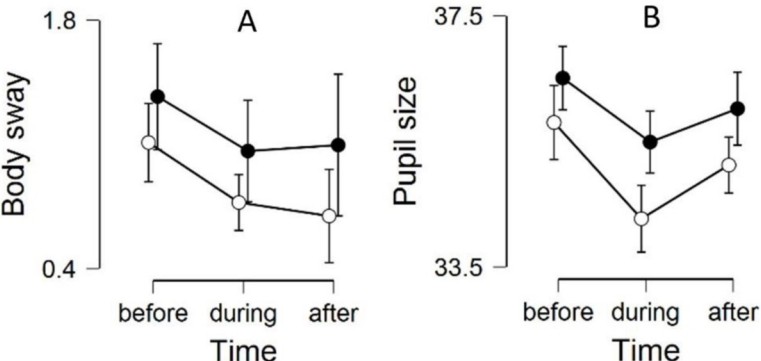

**Fig 4.** The response in A) body sway and B) pupil-size for high intensity (black) and low intensity (white) stimuli Before, During, and After the scene rotation. Values are presented as means with standard deviation.

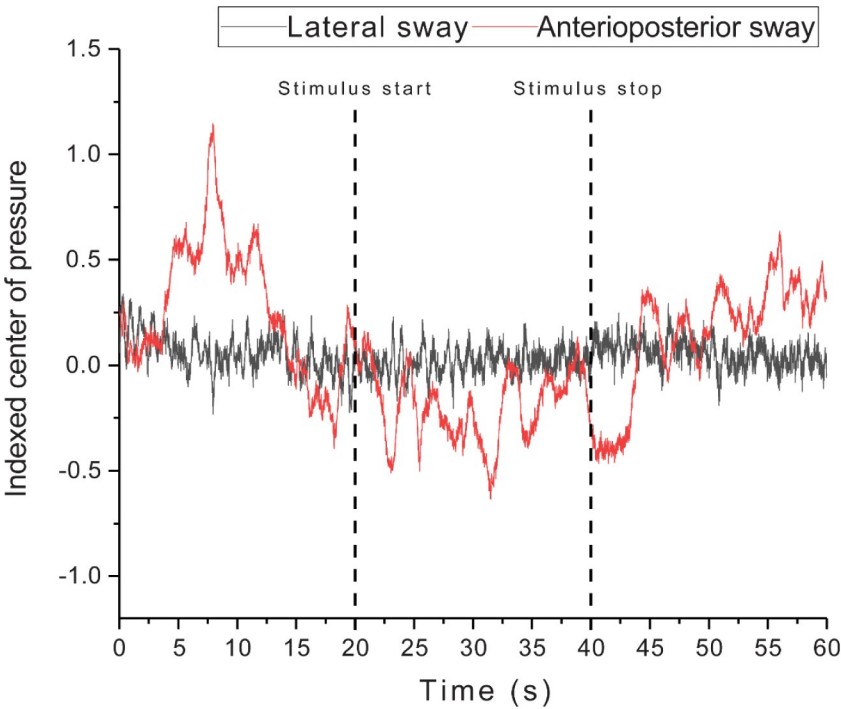

**Fig 5. Unmodified raw signal examples of body sway in the anteroposterior and lateral direction to high intensity stimulation.** The dashed line signifies stimulus start and stop.

The changes in pupil size can be seen in Fig 4. The Bayesian ANOVA for repeated measures strongly supported the main effects of Intensity and Time. There was also an interaction effect between the two variables (Intensity: $BF_{10}$ = 201.376; Time: $BF_{10}$ = 265.995; Intensity + Time: $BF_{10}$ = 363 627.085).

As for pupil activity, pupil oscillation amplitudes were sensitive to Time in a similar manner to both body sway area and pupil size, with the amplitude decreasing from Before, 1.42 px (0.59) to During 1.06 px (0.44) and then increasing to After again, 1.33 px (0.55). There was no difference between Intensities. The Bayesian ANOVA supported these results (Intensity: $BF_{10}$ = 0.990; Time: $BF_{10}$ = 171.576; Intensity + Time: $BF_{10}$ = 249.617). There was no significant relation between pupil oscillation frequency and other variables in neither the ANOVA nor the Bayesian ANOVA (Intensity: $BF_{10}$ = 1.991; Time: $BF_{10}$ = 0.170; Intensity + Time: $BF_{10}$ = 0.341).

## Subjective discomfort

Subjects' VAS score exhibited great spread between individuals, but were generally low. For the LI stimulus the median score was 0.2 (IQR 0.0–0.8), 0.1 (IQR 0.0–0.7) and 0.1 (IQR 0.0–0.6) for Before, During, and After respectively. The corresponding values for the HI stimulus were 0.2 (IQR 0.00–0.3), 0.4 (IQR 0.1–1.2) and 0.1 (IQR 0.0–0.4). There was no significant difference in VAS-score between neither Intensity nor Time, meaning that subjects' scored discomfort heterogeneously for both intensities over all time phases. No correlation analysis was done using VAS due to the obvious floor-effect of VAS scores.

## Correlations

A moderate correlation effect was found within the LI stimulation between ocular torsion and vertical skewing (r = 0.45, p = 0.04). Additionally, pupil size and torsional velocity were

moderately correlated (r = 0.49, p = 0.03), while the relation between pupil size and skewing velocity was weak, or non-existent (r = 0.04, p = 0.05) for LI. The HI stimulation yielded a moderate correlation between torsional velocity and body sway (r = 0.44, p = 0.04) as well as between torsional velocity and pupil size (r = 0.42, p = 0.05).

As for the pupillary response, averaged pupil size was strongly correlated to pupil oscillation amplitude for both LI (r = 0.79, p<0.001) and HI (r = 0.69, p<0.001). Oscillation amplitudes were in turn negatively and moderately correlated to oscillation frequency for both LI (r = -0.55, p = 0.01) and HI (r = -0.51, p = 0.02). Consequently, a larger pupil was associated with larger oscillations but also a slowing of the oscillation frequency.

## Discussion

The aim of this study was to investigate the effect of rotating visual clutter of different intensities on the eye movements of ocular torsion and vertical skewing, as well as changes in postural control, the sympathetic response, and the subjective sensation of discomfort during the trials. It was found that the level of visual clutter contained in the visual scene had a strong impact on ocular torsion, body sway and the pupillary response. The second aim was to correlate these results to one another, investigating possible objective measures that could hold clinical utility in assessing balance complaints of a visual nature. It was found that the torsional velocity was moderately correlated with skewing, body sway and pupillary response.

### Eye movement responses

The gradual fading of the eye movement response, particularly seen in the torsional velocity, is a well-known phenomenon to optokinetic stimulation [25]. What is more noteworthy is the impact of the visual clutter intensity on the torsional response, with the high intensity resulting in a higher velocity throughout the stimulation. It has been shown that an increase in optokinetic stimulation speed results in an increased torsional slow-phase velocity as the torsional position of the eyes aim to compensate for an increased frequency of a sinusoidal visual rotation [26]. Consequently, it would seem reasonable to believe that increasing the density of visual information is having a similar impact on the reflexive eye movement response. Considering that an increased optokinetic velocity will yield a stronger sensation of self-motion [27, 28], one can stipulate that the effect of visual information density holds a similar effect on the brain, which would explain the reflexive motor response.

This study clearly corroborates our previous finding that vertical skewing is physiologically seen to optokinetic rotation [11]. The skewing response's more irregular pattern to stimulation reveals a separate response pattern than its torsional counterpart. Moreover, the separate reactions to visual intensities suggest that the eye movements are associated with separate pathways. Another implication of the differences in eye movement response is that the torsional reflex is more sensitive to visual clutter than vertical skewing, further indicated by the latter's lower Bayesian score. Such a relationship would indicate that the torsional response is responsible for the previously described change in torsion-skewing ratio seen to different intensities in visual clutter [11]. Consequently, as long as the duration and amplitude of the optokinetic stimulation is sufficient, it may be adequate to focus on the torsional response when evaluating the effects of rotating visual clutter on patients suffering from visual vertigo as it appear more reliable than skewing over time.

A rather subtle increase in visual information, as shown with the low-to-high intensities implemented in this study, resulted in a significantly higher eye movement velocity, and it is well described that incongruences between stimuli and eye velocities contribute to a decreased dynamic visual acuity due to feedback error in the complex sensory mechanisms [29]. Put into

a context of hypersensitivity to visual motion, the symptoms are primarily caused by mismatched visual and vestibular input signals brought on by increased visual dependency [8]. Consequently it would be of interest investigating the eye movement responses of vertiginous patients describing symptoms to visual clutter. While this study does not test visual acuity, it is tempting to suggest that a change in eye movement velocities related to visual feedback errors may be of particular interest in patients with visual motion hypersensitivity.

Results showed that those exposed to the high intensity stimulus before the low intensity exhibited slower torsional velocities overall. In addition, the same group did not show any substantial difference between visual intensities for the Middle and Late time slots. Based on this it would seem that the stronger visual provocation resulted in inhibiting the torsional response. It is well established that motion habituation is reflected in the eye movement response [30]. However, the inhibition in torsional velocity seen here happened almost instantaneous, with little time for any habituation effect to manifest. The oculomotor response is nevertheless decreased, possibly by desensitization, and remained so even after the five minute waiting period between trials.

## Body sway and pupil size

While dynamic sampling rates are commonplace, the in-house developed WBB software yielded a rarely used frequency of 99.1 Hz. While this rate was established over several timed recordings, one must consider that inconsistent sampling during this interval was a possibility, posing a limitation to the body-sway measurement of this study. The effect on the analysis should however be minimal since the body sway area was calculated over a time period of 20 seconds. Similarly the concept of measuring luminosity to a moving scene is complicated. Measuring room luminosity does not reflect the light hitting the retina. Similarly, using a type of artificial fovea implemented in this study, measuring candela at specific time points, risks not being representative of a trial in its entirety. We aimed to minimize this risk by performing 15 measures for each trial. As no trend in altered luminosity was seen between the intensity levels or between their static or dynamic presentation, one can argue that the luminosity levels would not explain the significant and coherent changes seen during the study.

Body sway and pupil size decreased significantly during the stimulation phase for both intensities, which was unexpected considering that visual motion is known to induce both postural instability and affect autonomic responses [31]. In contrast, the similarities between body sway and pupil size were expected due to the pupil's known reaction to said autonomic changes [32]. Much like the torsional response, both body sway and pupil size were sensitive to visual intensity levels over the three time phases. Subjects exhibited decreased postural control accompanied with greater pupillary dilatation during the high intensity stimulation phase relative to the low intensity stimulus.

The visual stimulation was presented as a rotational movement around a central pivot point, simulating how retinal optic flow would be induced during a head tilt movement. As such, one could expect that body sway primarily would be affected in a lateral fashion, i.e. left-to-right or vice versa depending on stimulus direction. Surprisingly, this study revealed a largely intact sideways posture, with a larger anteroposterior (front-to-back) sway that tended to be more readily influenced by the visual rotation as illustrated in Fig 5. Such a finding may be explained by the much higher lateral stability associated with keeping one's feet in a straight lateral line at shoulder width; naturally, such a positions meant the postural foundation was greater in width rather than depth, as the feet are shorter than the distance between the shoulders.

The increased body sway area for the high intensity stimulation was in line with our hypothesis that an increased visual clutter would lead to decreased postural control. This is particularly well-established with patients suffering from visual vertigo, or visual motion hypersensitivity, and elderly who experience intensified balance impairment and subjective discomfort in the presence of visual clutter [8, 33]. However, both body sway and pupil size decreased during the stimulation phase, which was contrary to our hypothesis of a decreased balance stability to the optokinetic stimulation. While a decreased body sway over time may be explained by habituation to the task at hand, such a phenomenon would not explain the subsequent increase in body sway after the visual motion had ceased. Also, the controls viewing a stationary blank visual scene showed no such time-sensitive habituation.

While the balance provoking effects of visual clutter is usually related to motion, this study showed a sensitivity to visual intensity in terms of body sway and pupil size even during the initial 10 seconds of each trial when the visual scene was stationary. It is evident that a stationary visual field induces a sympathetic response relative to the level of clutter it contains. The difference in pupil size and body sway between intensities were however more manifest during the active visual rotation.

It is well-described that visual clutter can lead to an increase in performance distractors and perceptual errors when performing tasks [34]. This study suggests that some negative effects of visual clutter are present even in the absence of a task, as reflected in the brain's autonomic response during the Before period. Reducing visual clutter has been suggesting as a way of increasing well-being in the work place [35]. The results of this study hints at a possible biological mechanism behind such suggestions as stationary visual clutter increases both body sway and pupil size.

## Subjective discomfort

The reported discomfort was generally low, albeit heavily skewed within groups. This was expected given that all subjects were healthy and had no history of dizziness in addition to the uncertainties involved in using self-assessment forms for comparisons between groups. As expected, this study confirms that people subjectively rate their balance discomfort very differently, meaning that our hypothesis of higher density visual clutter causing increased discomfort could not be sustained under the current methodology. Considering that the question was posed in rather broad terms in relation to the sensation of discomfort, it is understandable that subjects interpret the form differently, limiting the use of the VAS in this study. This was especially true considering the relatively low number of participants. In future studies, questionnaires such as the Visual Vertigo Analogue Scale may be more apt in addressing the specific impact of visual clutter [36].

## Correlations

The purpose of the correlational analyses was to investigate the validity of using eye movement parameters as potential biomarkers for vertiginous patients. While the correlations between torsional and skewing eye movements during the low intensity stimulation was moderate, no such correlation was seen for the high intensity trials. This further indicates that the two motor outputs behave differently to visual clutter. As for pupil activity, the strong correlations between average pupil size, oscillation frequency and oscillation amplitude was expected as a higher frequency necessitates a lower amplitude.

The VAS-score was generally low and a clear floor-effect was found, which can be interpreted as a result of the difficulties using the VAS for grading balance discomfort in a healthy population. Additionally, the VAS score were not normally distributed, complicating any such

analysis of correlation to a normally distributed variable. It is possible that the VAS is more adequate to use in a population with symptoms relating to visually induced balance discomfort.

Pupil size had a moderate correlation with torsional velocity, but a weak, essentially non-existing, relation with skewing velocity. Consequently, it would seem that visual clutter is reflected in both pupil size and torsional velocity. This may be of value in that pupil size is generally easier to measure as more eye tracking systems allow for it. It could therefore be possible to use pupil size as a proxy for estimating the torsional response.

During high intensity visual stimulation, the correlation between torsion velocity and body sway was such that a greater torsional eye movement response was associated with poorer postural control. What stands out from the previous correlation of torsion and pupil size is that there was no significant correlation between body sway and torsion during the low intensity stimulation. It may be that body sway was easier to supress during the low intensity stimulation, as compared with the reflexive torsional response that lacks somatic innervation. Such a relationship is seen for patients suffering from VMH as increased visual clutter is associated with a decrease in postural control [8]. The relationship between body sway and torsional velocity may hold clinical utility in identifying objective variables for assessing these patients. Considering that the correlation between body sway and torsional velocity was stronger for the high intensity stimulation it is tempting to suggest that subjects who are more sensitive to visual motion may exhibit such a correlation already at lower intensity levels as they are more visually dependent [37].

## Conclusion

The results of this study indicate that visual clutter has a significant impact on the body. Subjects' body sway and pupillary response increased when viewing a higher intensity of visual clutter, both moving and stationary. This response could be of clinical value when assessing patients suffering from visual motion hypersensitivity. It could also explain why some healthy individuals feel discomfort from simply being in visually cluttered surroundings.

The velocity of the ocular torsion during the rotating visual clutter was significantly higher during the higher intensity, and moderately correlated to both body sway and pupil size. This could make it a potential candidate for evaluating how visual clutter may effect a patient's balance response without having to implement physically demanding balance-tasks. Furthermore, the pupillary response, being correlated to the torsional velocity, may prove a useful proxy for assessing visual balance complaints, as it is more readily traceable by eye-tracking software.

## Supporting information

**S1 Data.**
(PDF)

## Author Contributions

**Conceptualization:** Tobias Wibble, Tony Pansell.

**Formal analysis:** Tobias Wibble, Tony Pansell.

**Investigation:** Tobias Wibble, Ulrika Södergård.

**Methodology:** Tobias Wibble, Frank Träisk, Tony Pansell.

**Project administration:** Tobias Wibble.

**Resources:** Tony Pansell.

**Supervision:** Tobias Wibble, Frank Träisk, Tony Pansell.

**Validation:** Tony Pansell.

**Visualization:** Tobias Wibble.

**Writing – original draft:** Tobias Wibble, Ulrika Södergård.

**Writing – review & editing:** Tobias Wibble, Tony Pansell.

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
