## [Decision Letter · Decision Letter 0]

9 Oct 2019

PONE-D-19-19568

Intensified visual clutter induces increased sympathetic signalling, poorer postural control, and faster torsional eye movements during visual rotation

PLOS ONE

Dear Dr Wibble,

Thank you for submitting your manuscript to PLOS ONE. After careful consideration, we feel that it has merit but does not fully meet PLOS ONE’s publication criteria as it currently stands. Therefore, we invite you to submit a revised version of the manuscript that addresses the points raised during the review process.

We have received reviews from two reviewers who both believe that the manuscript has merit, but will benefit from further clarification.  In particular, ensure that sufficient detail is provided in the methods and that redundancy is limited.

We would appreciate receiving your revised manuscript by Nov 23 2019 11:59PM. To enhance the reproducibility of your results, we recommend that if applicable you deposit your laboratory protocols in protocols.io, where a protocol can be assigned its own identifier (DOI) such that it can be cited independently in the future. For instructions see: http://journals.plos.org/plosone/s/submission-guidelines#loc-laboratory-protocols

We look forward to receiving your revised manuscript.

Kind regards,

Eric R. Anson

Academic Editor

PLOS ONE

Journal Requirements:

1. We note that you have indicated that data from this study are available upon request. PLOS only allows data to be available upon request if there are legal or ethical restrictions on sharing data publicly. For more information on unacceptable data access restrictions, please see http://journals.plos.org/plosone/s/data-availability#loc-unacceptable-data-access-restrictions.

2. Please ensure that you refer to Figure 2 in your text as, if accepted, production will need this reference to link the reader to the figure.

Reviewers' comments:

Reviewer's Responses to Questions

**Comments to the Author**

1. Is the manuscript technically sound, and do the data support the conclusions?

Reviewer #1: Partly

Reviewer #2: No

2. Has the statistical analysis been performed appropriately and rigorously? 

Reviewer #1: Yes

Reviewer #2: No

3. Have the authors made all data underlying the findings in their manuscript fully available?

Reviewer #1: Yes

Reviewer #2: No

4. Is the manuscript presented in an intelligible fashion and written in standard English?

Reviewer #1: No

Reviewer #2: Yes

5. Review Comments to the Author

Reviewer #1: This study explored postural ocular and pupil responses to rotating visual clutter in 16 healthy participants. Two intensities of rotating visual clutter were used with the higher of the two intensities resulting in significantly greater torsional velocity, postural instability and pupil size. The paper suggests that these responses may suitable for clinical assessment of patients suffering from visual motion hypersensitivity.

Specific comments:

Abstract:

Line 18: Describe difference in intensity as later (line 23) you mention specific results for high intensity but have not described what the difference is.

Introduction:

Line 56: Define ocular balance-response, reference or define.

Line 58: Reword and describe “they are, however, also seen to rotational optokinetic stimulation.”

Line 73: Be wary of performing correlations between all variables. Ensure you have clear reason/hypotheses behind correlation performed.

Methods:

Line 80: Define or reference balance discomfort. Does this mean instability.

Line 81: “Subjects were balance in terms …”. In a paper investigating balance, maybe use the term “subjects were age and sex matched” to avoid confusion.

Line 81: “Subjects were balanced in terms of age and sex (4 men and 4 women aged 19 to 25, and 4 men and 4 women aged 49 to 65)”. Define the two groups and why do the ages not match even though you state they are?

Line 86: Were all canals assessed using HIT or just horizontal canals?

Line 93: Reword sentence

Figure 1: Legend and figure do not match. “A represents the high intensity stimulus (LI)” but figure 1A is a high intensity stimuli

Figure 1 legend: Correct “0,42cm”

Line 129: Provide further detail about calibration process

Line 134: “control that the head was immobilized”. Was the head actually immobilized or were participants instructed to not move the head?

Line 137: What was the specific criteria for rejection “noticeable movement”

Line 144: Why does the WBB sampling frequency vary? Do you analyze data using each trials specific sample frequency (i.e. for filtering purposes) or treat all trials as though they were sampled at 99.1 Hz

Line 159: Why was a blank visual field used and not a stationary version used during baseline in other trials?

Line 162: As discomfort was not defined, it becomes hard to interpret any results collected via VAS. Are participants reporting instability or motion sickness.

Line 196: ‘In the Origin software’?

Line 221: ‘Alfa’

Results:

Line 233: “in the compensatory direction” describe what this compensation is (i.e. visual rotation clockwise evoked ……..)

Table 1,2&3: Units

Figure 2 is not referenced in text

Figure 2&4: Left and right graphs could benefit from being label A and B to aid in-text referencing.

Figures 2,4&5: Units and what do error bars represent (STD or SEM).

Figure 5: correct y axis numbering (0.5 not 0,5 etc)

Line 300: units

Line 315: Correlation figures could be beneficial

Discussion:

Line 349-351: How??

Line 356: Reword sentence

Line 363-364: If skewing is purely a vestibular response. Shouldn’t you see no response as all as the head is stationary, thus no vestibular input?

Line 420: Does balance discomfort actually mean balance impairment/ instability.

Line 430: ‘even when the visual scene was stationary’ The visual scene was not stationary, or are you referring to the blank scene for the control group.

Line 431: reword

Line 440: I find it hard to understand this section as subjective discomfort is not defined, so unsure as to what participants were trying to portray through their answers. You use the word discomfort to describe postural instability in this manuscript and then in this section. I think it would be best to not use the word discomfort in the context of postural instability, thus potentially removing some confusion.

Line 462: As previously mentioned, correlations should be performed to examine hypothesis and relationships. It would appear that you performed correlation without clear reason ‘ as there is no clear reason for why this correlation would be seen’.

Reviewer #2: The authors measured how exposure to a display of rotating lines of two different densities affects ocular torsion, vertical skewing, body-sway, pupillary responses, and subjective feeling of discomfort. In addition, they analysed the correlation between these measures in order to explore potential clinical utility for assessing patients with visual motion hypersensitivity. This is certainly an important research topic and the manuscript is easy to read. However, I think the manuscript should be improved by clarifying the rationale for particular measurements taken and analyses performed (i.e., comparing ‘early’ vs ‘middle’ vs ‘late’ for some measures and before and during rotations for others) in the study. In addition, some important information is missing in the methods, such as the task or procedure.

In the introduction, it would be helpful if the authors could provide more background information from the literature about why the particular measurements were chosen.

First, ocular torsion and vertical skewing are introduced as part of VOR, however the current study used visual rotation instead of vestibular stimulations such as head/body rotation. What is the relationship between visually-induced ocular torsion and vestibular ocular torsion? Is the relationship expected from the mismatch model?

Second, body-sway and pupil responses were not introduced but only mentioned as other measurements. It would be helpful to provide more information about why measuring them, and how they would fit in the mismatch model. I’m not an expert in this field, but one relevant reference that comes to mind is, Lubeck, Bos, & Stins (2015), who showed that visual density of a rotation display affected body-sway. The paper by van Ombergen et al (2016) might also be relevant, as it describes the effect of optokinetic stimulation in vestibular mismatch patients (although they don’t assess eye movements).

Several important details and the rationale for performing the particular analyses are missing in the methods.

First, the procedure or task is not described. What were the instructions given to participants, especially regarding what they should do during the test? Were they sitting down or standing? Was fixation explicitly required?

Second, it is not clear why the statistical analyses were performed in this manner. The authors describe that the aim is to investigate the effect of rotating visual clutter (line 68), and the hypothesis is that higher intensity would result in stronger responses (line 74-76). Following this, I would expect the authors to compare responses during static and rotation stimuli, with intensity as the factor for ANOVA. It is not clear why the authors chose to use ‘Early, Middle and Late’ as time factors, and why eye movement responses and body-sway/pupil responses were analysed differently. It is also unclear whether the correlational analyses were corrected for multiple comparisons.

Lubeck AJA, Bos JE, Stins JF (2015) Interaction between Depth Order and Density Affects Vection and Postural Sway. PLoS ONE 10(12): e0144034.

Van Ombergen A, Lubeck AJ, Van Rompaey V, Maes LK, Stins JF, Van de Heyning PH, et al. (2016) The Effect of Optokinetic Stimulation on Perceptual and Postural Symptoms in Visual Vestibular Mismatch Patients. PLoS ONE 11(4): e0154528.

Some minor comments, mostly regarding clarification of procedure and analyses, or further discussion about interpretation are:

1. Line 81. It is hard to ignore the age gap between participants, which intuitively would have an effect on vestibular functions. Age should be considered as a factor in the analysis. If not, maybe it would be helpful to just use a few sentences to explain why.

2. Line 93. “The visual stimuli showed xxx”, should be “The visual stimuli are/is xxx”.

3. Line 100. What do the authors mean by ‘evenly randomized’? Was one participant exposed to one direction only?

4. Line 117. “0,42cm” should be “0.42 cm”.

5. Line 131. Were the 3D head movements also recorded by Chronos, or a different system was used? What was the system used?

6. Line 148. It is not entirely clear to me what the raw data from the WBB consists of, and how the authors processed these data. Also, given the differences in A-P and M-L body sway, the authors might want to consider treating these as separate variables. I would expect that the stimulus should only affect M-L sway, and the A-P sway is unrelated to the stimulus.

7. Line 154. Proprioception and muscular or joint fatigue are not the same. Please clarify the rationale, preferably with citations.

8. Line 174., Could the authors clarify how the raw data were processed? For example, what filters were used to process raw position data, and what criteria was used to separate quick phases and slow phases, and over which time intervals were each of the variables calculated?

9. Line 187. Related to the major points described above, why were three time intervals chosen, and why are there ‘breaks’ between the time intervals?

10. Line 191. It is not clear why the body sway area was calculated this way. Wouldn’t the vertical and horizontal positions produce a trajectory rather than a scatter plot? Is it common to fit an ellipse to define the body sway area?

11. Line 210. Should “between” should be “within” here?

12. Line 211. Were any post-hoc tests performed? If yes, what type?

13. Results. There is redundancy in the presentation of the data. Figures are more preferable than tables. Several tables are redundant, such as table 1 (contents already shown in figure 2) and table 3 (figure 4).

14. Line 247. Statistic test values are missing. What test was performed here?

15. Figure 2 is not mentioned in the text.

16. Figure 3. It would be helpful to show these raw data before the averaged data. There are some inconsistencies in the figure. For example, the y axis says “Amplitude (degrees)”, but one of the legends says “head velocity”. Line 329. The VAS shows a strong floor effect, and I therefore don’t think it is appropriate to include this measure in the correlation analysis.

17. Discussion. The authors should discuss the effects of stimulus intensity during the ‘before’ period. It would also be helpful to discuss the conflict between the hypothesis and results regarding body-sway, namely that body sway seems to decrease during rotation. Is this true for both A-P and M-L sway when these measures are analysed separately?

18. Discussion. Pupil sizes can be affected by many factors, and one possible confound in the current study could be luminance. I don’t think luminance of five points on the screen (line 109-111) can represent the average luminance of the display, since the visual stimuli are not uniform across the space. If the contrast of the line and the background is the same in both intensity conditions (line 103), then the average luminance of LI is higher than that of HI, which potentially could lead to smaller pupil sizes. This could also explain why the difference in pupil size exists even when the visual stimuli are static. The claim that pupil size can be an indicator for torsion (line 473-474) is therefore not convincing. Could the authors clarify or discuss this?

19. Discussion. It is not clear what the authors mean by “work load” (line 370-371). If participants were just passively viewing the screen, what is the “work load” required? Could the authors provide some evidence that increased visual information results in greater work load?

6. PLOS authors have the option to publish the peer review history of their article (what does this mean?). If published, this will include your full peer review and any attached files.

Reviewer #1: No

Reviewer #2: No

---

## [Author Response · Author response to Decision Letter 0]

15 Nov 2019

There are no restrictions on sharing the original data used in this study. We apologize for the oversight and have uploaded the data sheet with the revised manuscript.

---

## [Decision Letter · Decision Letter 1]

9 Dec 2019

PONE-D-19-19568R1

Intensified visual clutter induces increased sympathetic signalling, poorer postural control, and faster torsional eye movements during visual rotation

PLOS ONE

Dear Dr Wibble,

Thank you for submitting your manuscript to PLOS ONE. After careful consideration, we feel that it has merit but does not fully meet PLOS ONE’s publication criteria as it currently stands. Therefore, we invite you to submit a revised version of the manuscript that addresses the points raised during the review process.

This version of the manuscript is much improved, but as you can see from the comments below there are still some areas that will benefit from further clarification.  In particular, be more clear about the aims such that all non-exploratory analyses are linked to a testable hypothesis as this will enhance readability of the manuscript.

We would appreciate receiving your revised manuscript by Jan 23 2020 11:59PM. To enhance the reproducibility of your results, we recommend that if applicable you deposit your laboratory protocols in protocols.io, where a protocol can be assigned its own identifier (DOI) such that it can be cited independently in the future. For instructions see: http://journals.plos.org/plosone/s/submission-guidelines#loc-laboratory-protocols

We look forward to receiving your revised manuscript.

Kind regards,

Eric R. Anson

Academic Editor

PLOS ONE

Reviewers' comments:

Reviewer's Responses to Questions

**Comments to the Author**

1. If the authors have adequately addressed your comments raised in a previous round of review and you feel that this manuscript is now acceptable for publication, you may indicate that here to bypass the “Comments to the Author” section, enter your conflict of interest statement in the “Confidential to Editor” section, and submit your "Accept" recommendation.

Reviewer #2: (No Response)

2. Is the manuscript technically sound, and do the data support the conclusions?

Reviewer #2: Partly

3. Has the statistical analysis been performed appropriately and rigorously? 

Reviewer #2: Yes

4. Have the authors made all data underlying the findings in their manuscript fully available?

Reviewer #2: Yes

5. Is the manuscript presented in an intelligible fashion and written in standard English?

Reviewer #2: Yes

6. Review Comments to the Author

Reviewer #2: I am pleased with the additions that the authors made in the introduction and methods, it very much improved the readability of the manuscript.

However, I still find the aims (in the introduction and discussion), hypothesis and corresponding data analysis somewhat messy. I think it is important to clearly define the aim, and formulate hypotheses and analyze the data in line with this aim. The current (first) aim is:

“to investigate the effect of rotating visual clutter on a series of variables in healthy adults: 1) The oculomotor response of ocular torsion and vertical skewing, 2) the postural response, measured as body-sway, 3) the sympathetic response, evaluated through monitoring pupil size, and 4) the subjective sensation of discomfort, evaluated through a subjectively self-reported visual analogue scale (VAS).”

However, the authors then look at the effect of the density of the stimulus and at the time course of oculomotor responses during the viewing of rotating visual clutter. For body sway, pupillary response, and subjective discomfort, the authors look at the effect of the rotation and density of the stimulus (the hypothesis concerns the effect of density). I would mention the intensity in the aim and separately describe the aim for the oculomotor responses. Also, in the discussion (line 403-405) the authors write “It was found that the level of visual clutter contained in the visual scene had a strong impact on ocular torsion, body-sway and the pupillary response over time”. However, the time course was only analyzed for torsion and skewing. Please be consistent in the description of aims, hypotheses, and analyses.

Furthermore, did the authors consider analyzing the full time course of the pupillary response (like torsion and skewing), since this is also a continuous variable? Finally, if there are differences between pupil sizes of younger and older people, this is a confound in the correlation analyses. Scatter plots for the correlations would be a nice addition.

I also have a few minor comments. Line numbers refer to the version in which the changes are highlighted.

Minor comments:

Previous comment 9: Why were three intervals chosen as opposed to, for example, two or four intervals? The choice seems arbitrary. If the authors are interested in examining the time course, why not analyze the full time course of the slow phase movements?

Previous comment 14: This comment refers to the sentences on line 247-250 of the original manuscript: “… there was no significant difference between the Middle and Late period, while the Early response was the highest. There was no significant effect of Intensity or Intensity order for the stimulation period.” Were t-tests performed for comparing Early, Middle and Late? Please provide the test values for these statistical tests.

Previous comment 16: I’m sorry that this comment was unclear. I intended to suggest to present the raw data before the average data, that is, to swop figure 2 and 3.

Line 66-67: What would be the use of a biomarker to identify vertigo? If the symptoms are present, do we need a biomarker?

Line 69-72: “We have … increased clutter.” This sentence is grammatically incorrect. Please correct to: “In a recent study, we have shown … per degree of skewing in response to increased clutter.”

Line 78: A word is missing after “proxy”.

Line 210-215: “Additionally … intensity level.” I don’t fully understand what the authors have written here, please clarify.

Figure 2: This is not critical, but it would be great if the authors could make a plot of the full time course of these variables during the 20 s of rotation for the LI and HI condition (i.e., select the slow phases and remove the quick phases, average across trials within each condition, then average across participants).

Line 340: Please remove the sentence referring to table 3.

Line 393-398: I apologize that I missed this in the previous round, but why is oscillation amplitude introduced here?

Line 438-449: I still don’t think the authors’ reasoning here is correct. Neurons can show tuning for visual spatial frequency or (rotational) speed but increased stimulation does not simply result in an increased signal, or increased ‘work’ for the brain. Please revise or provide references.

7. PLOS authors have the option to publish the peer review history of their article (what does this mean?). If published, this will include your full peer review and any attached files.

Reviewer #2: No

---

## [Author Response · Author response to Decision Letter 1]

11 Dec 2019

Dear Dr. Anson,

Thank you for the valuable suggestions provided in regards to our submission “Intensified visual clutter induces increased sympathetic signalling, poorer postural control, and faster torsional eye movements during visual rotation”. We are pleased to hear that our first revision was well received, and we consider the additional comments provided to be well presented in relation to the purpose of the study. After additional discussions within the author group we have performed additional revision to our manuscript to further clarify important key points. As per your request we have clarified the aims of the manuscript and presented the related hypothesis in a more structured manner, and highlighted the explorative nature of the correlations analysis for increased readability. 

We present our point-by-point response with regards to the original feedback, and what changes we have made. The comments are organized in relation to respective reviewer, with our answer highlighted in blue. Changes in the manuscript have also been highlighted in blue, with line-references added in this letter for your convenience. In some cases we have performed general revision, and as such cannot point to any specific point in the manuscript. These changes are also highlighted in blue so as to be easily identified. 

Thank you for your feedback and we look forward to hearing from you regarding our revision,

Tobias Wibble

---

## [Editor Report · Decision Letter 2]

18 Dec 2019

Intensified visual clutter induces increased sympathetic signalling, poorer postural control, and faster torsional eye movements during visual rotation

PONE-D-19-19568R2

Dear Dr. Wibble,

We are pleased to inform you that your manuscript has been judged scientifically suitable for publication and will be formally accepted for publication once it complies with all outstanding technical requirements.

With kind regards,

Eric R. Anson

Academic Editor

PLOS ONE

---

## [Editor Report · Acceptance letter]

19 Dec 2019

PONE-D-19-19568R2 

Intensified visual clutter induces increased sympathetic signalling, poorer postural control, and faster torsional eye movements during visual rotation 

Dear Dr. Wibble:

I am pleased to inform you that your manuscript has been deemed suitable for publication in PLOS ONE. Congratulations! Your manuscript is now with our production department. 

With kind regards,

on behalf of

Dr. Eric R. Anson 

Academic Editor

PLOS ONE